

# A new cropland area database by country circa 2020

Francesco N. Tubiello[1*], Giulia Conchedda[1], Leon Casse[1], Pengyu Hao[2], Giorgia de Santis[1] and Zhongxin Chen[2]

[1]Statistics Division, Food and Agriculture Organization of the United Nations, Viale delle Terme di Caracalla, Rome, Italy.
[2]Digitalization and Informatics Division, Food and Agriculture Organization of the United Nations, Viale delle Terme di Caracalla, Rome, Italy.

*Corresponding author:* francesco.tubiello@fao.org

## Abstract

We describe a new dataset of cropland area circa the year 2020, with global coverage, with data for 221 countries and territories and 34 regional aggregates. Data are generated from geospatial information on the agreement-disagreement characteristics of six open access high-resolution cropland maps derived from
remote sensing. The cropland area mapping dataset (CAM) provides information on: i) mean cropland area and its uncertainty; ii) cropland area by six distinct cropland agreement classes; and iii) cropland area by specific combinations of underlying land cover product. The data indicated that world cropland area is 1500 ± 400 million hectares (Mha) (mean and 95% confidence interval), with a relative uncertainty of 25% that increased across regions. It was 50% in Central Asia (40 ± 20 Mha), South America (180 ± 80 Mha), and
Southern Europe (40 ± 20 Mha); up to 40% in Australia and New Zealand (50 ± 20 Mha), South-eastern Asia (80 ± 30 Mha) and Southern Africa (16 ± 6 Mha). Conversely, cropland area was estimated with better precision, i.e. smaller uncertainties in the range 10%-25% in Southern Asia (230 ± 30 Mha), Northern America (200 ± 40 Mha), Northern Africa (40 ± 10 Mha), Eastern and Western Europe (40 ± 10 Mha). The new data can be used to investigate coherence of information across the six underlying products, as well as
to explore important disagreement features. Overall, 70% or more of the estimated mean cropland area globally and by region corresponded to good agreement of underlying land cover maps–four or more. Conversely, in Africa cropland area estimates found significant disagreement, highlighting mapping difficulties in complex landscapes. Finally, the new cropland area data were consistent with FAOSTAT in 15 out of 18 world regions, and for 114 out of 182 countries with a cropland area above 10 kha. By helping
to highlight features of cropland characteristics and underlying causes for agreement/disagreement across land cover products, the CAM dataset can be used as a tool to assess quality of country statistics and help guide future mapping efforts towards improved agricultural monitoring. Data are publicly available at: https://doi.org/10.5281/zenodo.7987515 (Tubiello et al., 2023a).




## 1 Introduction

Information on cropland area is needed to assess and monitor the sustainability of agriculture at local, regional and planetary scales. Information on world cropland area with national or sub-national detail is currently available as: i) Statistics of agricultural land use, collected from countries by the Food and

Agriculture Organizations of the United Nations (FAO) and disseminated in FAOSTAT (FAO, 2023a); and ii) land cover maps produced from remote sensing (Potapov et al., 2022a). These historically rather distinct sources are becoming interconnected, with remote sensed data increasingly complementing more traditional data sources such as agricultural censuses and surveys (Miller et al., 2009; Bailey and Boryan, 2010; FAO, 2018; Karthikeyan et al., 2020; Weiss et al., 2020; Bey et al., 2016). Comparison analyses at multiple scales

of these different sources have been published to facilitate in-depth understanding of cropland characteristics and to derive methods for data selection and applications (Bratic et al, 2019; Liu et al., 2021; Venter et al., 2022; Chaaban et al., 2022; Ding et al., 2022). We recently conducted a meta-analysis of the currently available six independent high resolution (10–30 m) land cover maps circa 2020 and derived a map on cropland agreement/disagreement at pixel level, showing that by combining such information world

cropland area can be estimated to within 25% (Tubiello et al., 2023b). That study identified 'definitional bias,' i.e., systematic errors due to imperfectly aligned land cover/land use definitions, as an important source of uncertainty in addition to well-described factors such as differences in data sources, pre-processing methods and validation approaches (Fritz et al., 2013; Gao et al., 2020, Wang et al., 2019). The cropland agreement maps are already being used in support of relevant geospatial work (e.g., Tang et al.,

55 2023).

This study presents a new database of cropland area at country level, based on the geospatial work of Tubiello et al. (2023b). We aggregate pixel level information and quantify means and uncertainties of cropland area at country and regional level. The new database, referred to hereafter as Cropland Agreement Mapping (CAM) dataset, provides information on cropland area by country, with information on: i) mean

estimate and uncertainty; ii) contribution to total area by agreement class; iii) contribution by specific combinations of the underlying land cover products.

The information provided by CAM, presented for the first time in this paper, helps to better understand the linkage between agricultural land cover and land use information and related uncertainty, offering useful insights with regards to future mapping efforts and their evaluation. The CAM dataset is available as open

access data at: https://doi.org/10.5281/zenodo.7987515 (Tubiello et al., 2023a).



## 2 Materials and Methods

### 2.1 Cropland agreement map

We used the cropland agreement map by Tubiello et al. (2023b) as input to generate country statistics. CAM consolidates information from six high-resolution land cover maps based on a meta-analysis done within the code editor of the Google Earth Engine (GEE) (Gorelick et al., 2017). These six maps are: ESRI (Karra et al. 2021); FROM_GLC Plus (Yu et al., 2022); GLAD (Potapov et al, 2022a); GLC-FCS30-2020 (Zhang et al., 2021); Globeland30 (Chen et al., 2015); and WorldCover (Zanaga et al., 2021). Appendix A further provides details on the characteristics of the six land cover maps, along with their spatial consistency and similarity analysis.

The cropland agreement map consists of two geospatial layers prepared at 30 m resolution as 8-bit unsigned integers: i) a *simple cropland agreement* map; and ii) a *detailed cropland agreement* map.

The *simple cropland agreement* map layer combines six cropland binary masks (with values of 1 for cropland, 0 for no cropland) from the six input land cover products (Appendix A), into a map with pixel values ranging 1-6, representing, when normalised by the number of layers, the probability of cropland area in each pixel.

The *detailed cropland agreement map* layer contains information on the individual land cover products and their combinations, with values ranging between 0 (bit 00000000, corresponding to no cropland), and 63 (bit 00111111, representing complete agreement) (Appendix B, Tab. B1). Each input dataset contains omission and commission errors, which affect their accuracy (Tab. 1). While the uncertainty information in CAM and in our dataset is computed net of these accuracies, it should be noted that by combining multiple land cover products, overall omission and commission errors may be reduced.

As discussed elsewhere (Tubiello et al., 2023b), the definitions of 'cropland' as a land cover class varied across the six products (Tab. 1), although they largely corresponded to FAO land use class *cropland* or *arable land* (Tab. 2). Specifically, of the six products used as input, GLAD, WorldCover and ESRI 'cropland' classes could conceptually be mapped to FAO land use class *arable land* or *temporary crops*, while the other three maps, including information on shrubs and woody components, could be better aligned with the FAO parent class *cropland* (Tubiello et al., 2023b). Within the latter, Globeland30 included within 'cropland' tree and shrub crops directly under a single class (cultivated land); FROM-GLC included permanent shrubs crops while excluding tree crops; and FCS30 provided data on cropland globally, with some partial regional distinctions between herbaceous and woody crops within irrigated/rainfed sub-classes (Tab. 1). The latter category was excluded to reduce definitional bias in the consolidated product.

**Table 1.** Cropland definitions and accuracy of the six input layers used for the cropland agreement map.

| Dataset | Label | Definition | Cropland class # | Accuracy[a] |
|---|---|---|---|---|
| **ESRI** (ESR) | Crops | Human planted/plotted cereals, grasses, and crops not at tree height; examples: corn, wheat, soy, fallow plots of structured land. | 5 | PA 89.9%; UA 91% |
| **FROM-GLC Plus**[b] (FRG) | Croplands | Land that has clear traits of intensive human activity. It varies a lot from bare field, seeding, crop growing to harvesting. It includes arable and tillage land with herbaceous/shrub crops and land with plastic foam or grass roof protection with distinguishing spectral properties. Fruit trees are classified into forests. | 10 – Level 1 | OA 71.9% |
| **GLAD** (GLD) | Cropland | Land used for annual and perennial herbaceous crops for human consumption, forage (including hay) and biofuel. Perennial woody crops, permanent pastures and shifting cultivation are excluded from the definition. The fallow length is limited to 4 years for the cropland class. | 1 | PA 86.4%; UA: 88.5% |
| **GLC-FCS30-2020**[c] (FCS30) | Cropland | Rainfed cropland, Irrigated cropland<br><br>Herbaceous cover<br>Tree or shrub cover (Orchard) | 10 – Level 1<br>20 – Level 1<br><br>11 – Level 2<br>12 – Level 2[d] | PA 88.0%; UA 83.9% |
| **Globeland30** (GL30) | Cultivated land | Category includes paddy fields, irrigated dry land, rain-fed dry land, vegetable land, pasture planting land, greenhouse land, land mainly for planting crops with fruit trees and other economic trees, as well as tea gardens, coffee gardens and other shrubs. | 10 | OA 85.7% |
| **WorldCover** (WCO) | Cropland | Land covered with annual cropland that is sowed/planted and harvestable at least once within the 12 months after the sowing/planting date. The annual cropland produces an herbaceous cover and is sometimes combined with some tree or woody vegetation. Note that perennial woody crops will be classified as the appropriate tree cover or shrub land cover type. Greenhouses are considered as built-up. | 40 | PA 76.7%; UA 81.1% |

[a] When available, user and producer accuracy (UA and PA) of the cropland class, overall map accuracy (OA) is otherwise reported. [b] Accuracy results reported for the 2020 map from the data producers (personal communication). [c] Accuracy results based on the 2015 version of the map. [d] Excluded from the cropland agreement map.

100

105



## 2.2 Preparation of the CAM dataset

The country statistics populating the CAM dataset (Tubiello et al., 2023a) were extracted from the simple and detailed agreement maps discussed above (Tubiello et al., 2023b), using the FAO Global Administrative Unit layer (GAUL) for country boundaries (FAO 2015)—also accessible from the GEE Code editor.

Generally, for $n$ land cover maps, pixel values in the *simple agreement map* belong to a set of $n+1$ elements, $\{0, 1/n, 2/n \ldots, (n-1)/n, 1\}$, representing the level of agreement among input maps, which was interpreted as the probability of finding cropland in each pixel by Tubiello et al. (2023b). By aggregating the pixel-level information at national scale it was therefore possible to generate country estimates of: i) mean cropland area $A$ and associated uncertainty: ii) cropland area by agreement class, $SA_1$ to $SA_6$, with $SA_k$ representing the area where $k$ maps agreed, and $A = \Sigma_k SA_k$ (Fig. 1); and iii) cropland area by specific map combinations.

Following the steps above, we generated values for over 221 countries and 34 territories, with country codes aligned to standard M49 area codes classification.

In terms of simple agreement classes, $SA_1$ represents the contribution to cropland area by single land cover products, while $SA_6$ provides information on the contribution to cropland area when all six products agree. Conversely, Similarly, for agreement class $SA_k$, data represent contributions to total cropland area by all relevant combinations of $k$ products. With respect to the two extreme cases, definitional bias would be small for $SA_6$, with CAM likely estimating subcomponents *temporary crops* rather than *cropland*, since the former category would be the only one that could be detected by all land cover products. By the same token, definitional bias would be higher for $SA_k$ classes and highest for $SA_1$, with CAM including in such cases more of those components of cropland that would not be equally detected by the underlying land cover products, for instance *temporary meadows and pastures* or *permanent crops*. To this end, CAM data on detailed cropland agreement provide the additional information on which specific combinations of $k$ land cover products contributed to class $SA_k$ in a given country. For example, the area of $SA_3$ in country $i$ could be the sum of agreement areas identified respectively by GLAD-WorldCover-ESRI and GLAD-WorldCover-FROM_GLC.


**Figure 1.** Simple cropland agreement map. Adapted from Tubiello et al. (2023b).

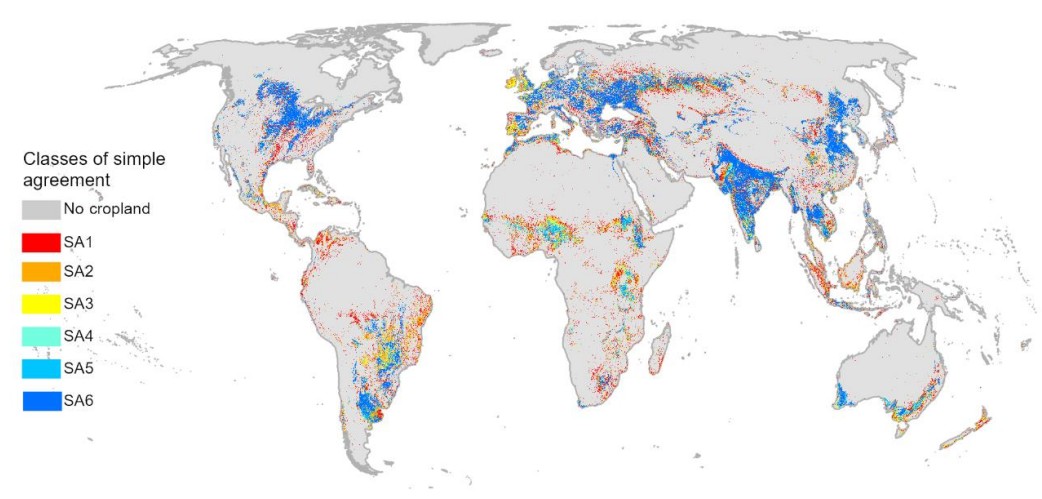


## 2.3 Comparison with FAO land use statistics

FAO land use statistics of cropland and arable land are routinely used as benchmark to assess the robustness

of land cover information at various scales (Vancutsem et al., 2013; Yu et al., 2014; Pérez-Hoyos et al., 2017; Xu et al., 2019; Li and Xu., 2020; Potapov et al., 2022a). We compared CAM cropland area estimates against FAOSTAT (FAO, 2023a) area statistics of *cropland* (Tab. 2), quantifying goodness of fit in terms of the Pearson correlation coefficient $R^2$ and a normalized root-mean square error (NRMSE, expressed in % and computed by dividing RMSE by the range of values). Possibly due to variations in land and water

masks, many small island states were absent in one or more of the six land cover inputs used in CAM. To ensure consistency, a minimum cut-off value of 10 thousand hectares (kha) of cropland was applied, resulting in 182 complete country records in CAM, out of the total 221 countries covered.




**Table 2.** FAO land use categories for cropland as defined in the FAO Land Use, Irrigation and Agricultural Practices questionnaire (FAO, 2023a).

| Land use category | Definition |
|---|---|
| **Cropland** | Land used for cultivation of crops. The total of areas under *Arable land* and *Permanent crops*. |
| *Arable land* | Land used for cultivation of crops in rotation with fallow, meadows and pastures within cycles of up to five years. The total of areas under *Temporary crops*; *Temporary meadows and pastures*; and *Temporary fallow*. *Arable land* does not include land that is potentially cultivable but is not cultivated. |
| *Temporary crops* | Land used for crops with a less-than-one-year growing cycle, which must be newly sown or planted for further production after the harvest. Some crops that remain in the field for more than one year may also be considered as temporary crops e.g., asparagus, strawberries, pineapples, bananas and sugar cane. Multiple-cropped areas are counted only once. |
| *Temporary fallow* | Land that is not seeded for one or more growing seasons. The maximum idle period is usually less than five years. This land may be in the form sown for the exclusive production of green manure. Land remaining fallow for too long may acquire characteristics requiring it to be reclassified, as for instance *Permanent meadows and pastures*, if used for grazing or haying. |
| *Temporary meadows and pastures* | Land temporarily cultivated with herbaceous forage crops for mowing or pasture, as part of crop rotation periods of less than five years. |
| *Permanent crops* | Land cultivated with long-term crops which do not have to be replanted for several years (such as cocoa and coffee), land under trees and shrubs producing flowers (such as roses and jasmine), and nurseries (except those for forest trees, which should be classified under "Forestry"). Permanent meadows and pastures are excluded from permanent crops. |

## 2.4 Communication of uncertainty and use of significant figures

The data made available through CAM are area estimates based on measurement samples of dimension $n=6$ (one value per land cover map), repeated over hundreds of pixels within a country. The true population mean cropland area $A$ was thus estimated via the sample mean area $x$ and its uncertainty $u$ as:

$$A = x \pm u \qquad (1)$$

Where $u = k*S/\sqrt{n}$ represented the 95% confidence interval, computed from the estimated standard error,

$S/\sqrt{n}$, $S$ being the sample standard deviation, multiplied by the factor $k=2.57$—corresponding to the two-tailed t-Student distribution cut-off value $t_{v,0.025}$ for $n-1 = 5$ degrees of freedom. We note that standard errors estimated from a sample of dimension $n$ carry a relative uncertainty of $1/\sqrt{(2n-1)}$, hence about 30% for $n=6$. This implies that $u$ should be communicated with one significant figure only (JCGM, 2009), affecting the communication of the estimated cropland area $A$ beyond the mere algebraic outcome of (1). To give an

example, estimates of world cropland area were expressed as $1500 \pm 400$ million hectares (Mha), even though the algebra in (1) yielded an apparently more precise result of $1540 \pm 370$ Mha. All results below were communicated accordingly.





## 3 Results and discussion

### 3.1 Cropland area

**3.1.1 Global and regional results**

CAM data indicated a total world cropland area in year 2020 of $1500 \pm 400$ million hectares (Mha), with a relative uncertainty of 27%. Uncertainty was higher across regions—up to 50% for Africa, Americas, Asia, Europe and 40% for Oceania (Tab. 3). The world's cropland area was very close to, and statistically consistent with the corresponding FAOSTAT value (1560 Mha). Comparisons of regional cropland area were also largely consistent with FAOSTAT, with $R^2 = 0.92$ and NRMSE of 8% (Fig. 2). With reference to Tab. 3, of the 18 world sub-regions considered, only three cropland area estimates were statistically inconsistent with FAOSTAT, namely Middle Africa, where we computed $16 \pm 7$ Mha vs. 37 Mha in FAOSTAT; Western Africa, $60 \pm 20$ Mha vs. 102 Mha in FAOSTAT; and South-eastern Asia, $80 \pm 30$ M ha vs. 123 M ha in FAOSTAT. All other fifteen sub-regional estimates had uncertainty bounds that
contained the corresponding FAOSTAT values.

The uncertainty of CAM values was higher at regional compared to world level, i.e., up to 50% in Central Asia, South America and Southern Europe; up to 40% in Australia and New Zealand, South-eastern Asia and Southern Africa (Tab. 3). In absolute terms, South America had in addition the largest absolute uncertainty (80 Mha). Conversely, cropland area estimates with the smallest uncertainties (hence larger
precision) were those for Southern Asia (13%), Northern America (20%), Northern Africa (24%), Eastern and Western Europe (25%).







**Table 3.** Regional cropland area estimates in CAM (means and uncertainties) and FAOSTAT. Melanesia, Micronesia, and Polynesia were excluded due to the the cut-off country size used in this study.

| Region | CAM | u | %u | FAO |
|---|---|---|---|---|
| | Mha | Mha | % | Mha |
| Eastern Africa | 70 | 20 | 29% | 78 |
| Northern Africa | 40 | 10 | 25% | 50 |
| Southern Africa | 16 | 6 | 38% | 14 |
| Western Africa | 60 | 20 | 33% | 102 |
| Middle Africa | 16 | 7 | 44% | 37 |
| | | | | |
| Northern America | 200 | 40 | 20% | 199 |
| Central America and Caribbean | 30 | 15 | 50% | 37 |
| South America | 180 | 80 | 44% | 131 |
| | | | | |
| Central Asia | 40 | 20 | 50% | 39 |
| Eastern Asia | 170 | 40 | 24% | 145 |
| Southern Asia | 230 | 30 | 13% | 240 |
| South-eastern Asia | 80 | 30 | 38% | 123 |
| Western Asia | 50 | 10 | 20% | 44 |
| | | | | |
| Eastern Europe | 200 | 50 | 25% | 197 |
| Northern Europe | 30 | 10 | 33% | 19 |
| Southern Europe | 40 | 20 | 50% | 37 |
| Western Europe | 40 | 10 | 25% | 35 |
| | | | | |
| Australia and New Zealand | 50 | 20 | 40% | 32 |
| | | | | |
| World | 1500 | 400 | 27% | 1560 |




Data

**Figure 2.** Regional comparisons between CAM and FAOSTAT data ($R^2$=0.92; NRME= 8%).

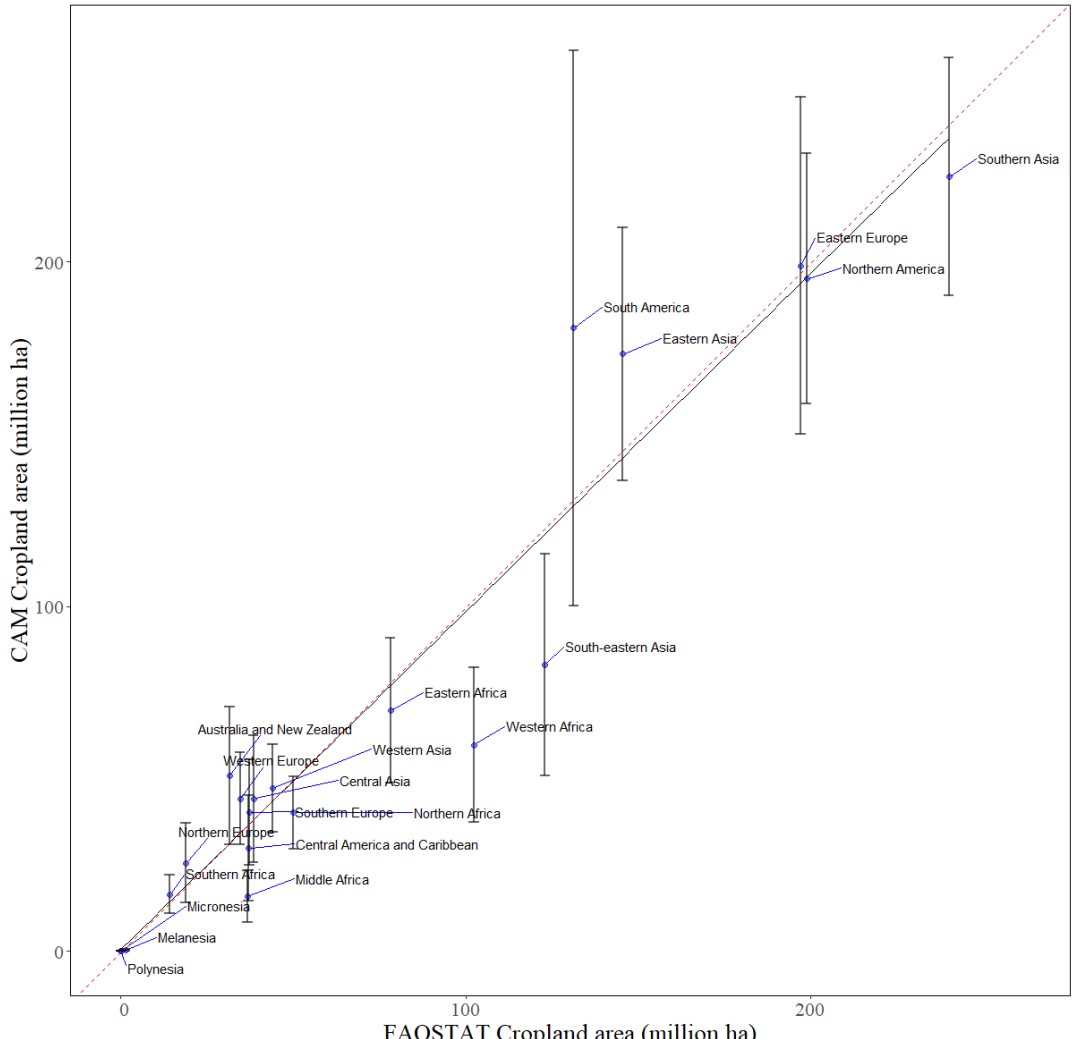

### 3.1.2 Country results

CAM estimates compared well with FAOSTAT statistics also at country level. Overall, considering the 182 countries and territories with estimated cropland area greater than 10 kha, CAM values were in good agreement with FAOSTAT data ($R^2$ = 0.95; NRMSE of 3%, or less than 5 Mha on average). In addition, CAM estimates were statistically consistent with FAOSTAT values for 114 of the 182 countries considered (Fig. 3). Conversely, among CAM estimates that were inconsistent with FAOSTAT data, relevant cases



(cropland area > 1 Mha) were Colombia (4 ± 3 vs. 9 Mha); Côte d'Ivoire (3 ± 2 vs. 8 Mha); the Democratic Republic of Congo (DRC) (5 ± 5 vs. 15 Mha); Indonesia (20 ± 10 vs. 59 Mha); Malaysia (3 ± 3 vs. 8 Mha); Niger (5 ± 4 vs. 18 Mha); Pakistan (21 ± 3 vs. 32 Mha); and the Philippines (6 ± 3 vs. 11 Mha). These highlight the need for further investigation of land cover maps and FAO statistics to better identify possible data quality issues.

**Figure 3.** Country comparisons between CAM and FAOSTAT ($R^2$=0.95; NRME= 3%).

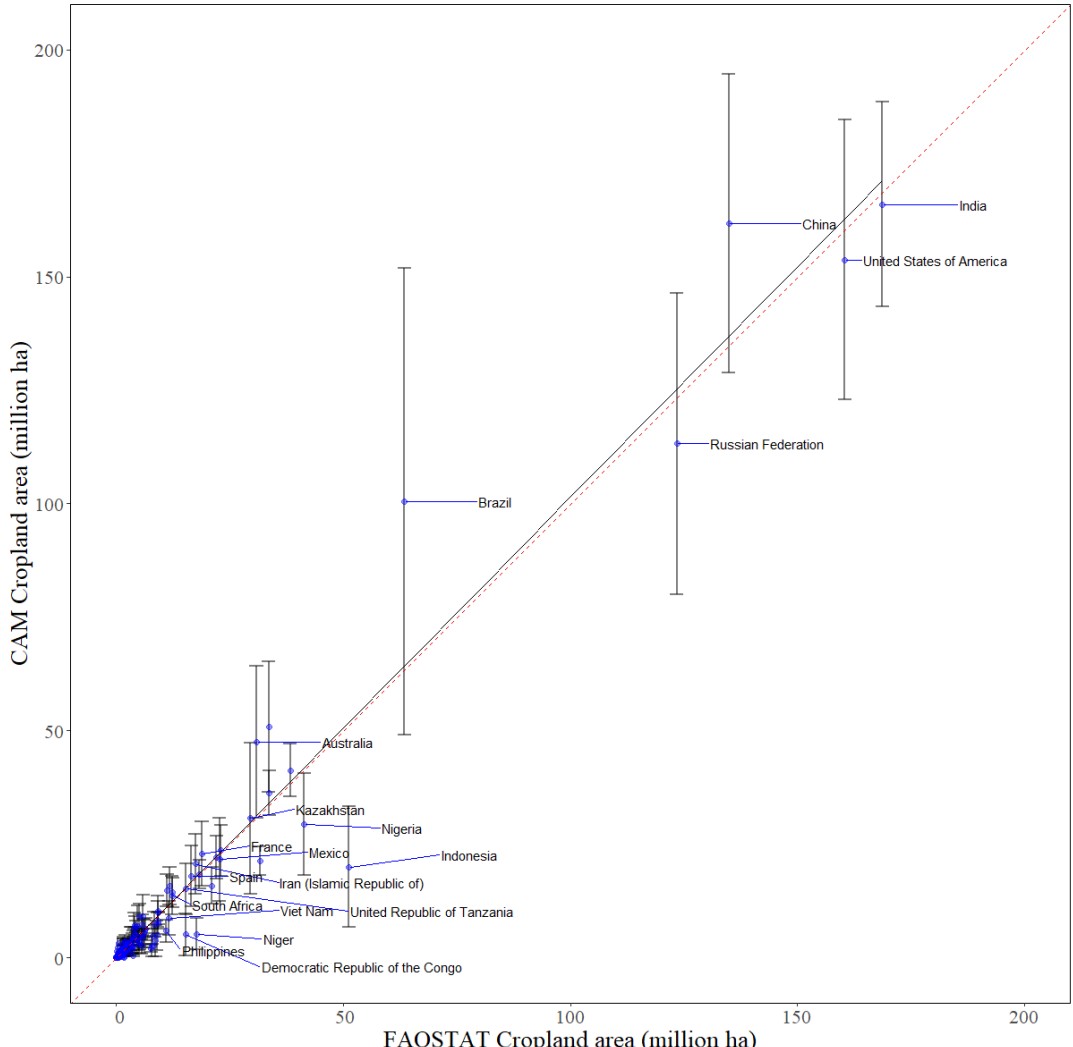



Overall, the large range of uncertainties found in CAM country data, 20%-100%, underscored a large variability across geographies, in relation to: i) complexity of cropland landscapes; and/or ii) the ability of single land cover products to capture them consistently across regions. We note nonetheless that among the top five countries in terms of cropland area extent (i.e., Brazil, USA, China, India, Russian Federation), only Brazil showed high uncertainty (50%) (Fig. 3). Countries with estimated cropland area greater than 1 Mha and relative uncertainties of 100% included Malaysia (3 Mha in absolute value), Nicaragua (2 Mha); Ireland (2 Mha); New Zealand (3 Mha); and DRC (5 Mha) (see also Appendix C).

## 3.2 Simple cropland agreement

### 3.2.1 Regional results

CAM data complement the information on cropland area with knowledge of the contribution by agreement class. This information added useful insights into some of the findings highlighted above. First, the data show that the top agreement class, $SA_6$ was in general the larger contributor to the estimated cropland area at regional level, with the exception of Africa (Fig. 4). This indicated that the underlying land cover maps were capable to map cropland rather consistently in most regions. More specifically, the sub-regions with the highest contribution of agreement class $SA_6$ (>50%) were: Northern America ($SA_6 > 62\%$), Eastern Europe ($SA_6 > 58\%$) Western Europe ($SA_6 > 56\%$) and Southern Asia ($SA_6 > 56\%$) (Appendix C, Tab. C1).

The large contribution of $SA_6$ was consistent with prevalence of simpler cropland landscapes in those regions—for instance, landscapes characterised by large fields with high-input annual crops. Conversely, the lowest contributions of $SA_6$ to cropland area were estimated in Middle Africa (2%), Western Africa (5%) and Eastern Africa (10%), followed by regions with shares of a quarter to a third, that is Central America (25%), Southern Africa (28%), Northern Africa (29%), Central Asia (32%), Australia and New Zealand (33%). By the same reasoning as above, low shares of $SA_6$ were indicative of regions with prevalence of more complex agricultural mosaics—including in particular more traditional low-input systems. The latter characterize agriculture in Eastern, Middle and Western Africa regions. CAM data show that, unlike all other regions, in all three the $SA_6$ contribution to cropland area was not only in the single digits, but far smaller than contributions from other agreement classes (Appendix C, Tab. C1).



**Figure 4.** Percentage contribution to cropland area by simple cropland agreement, by subregion.

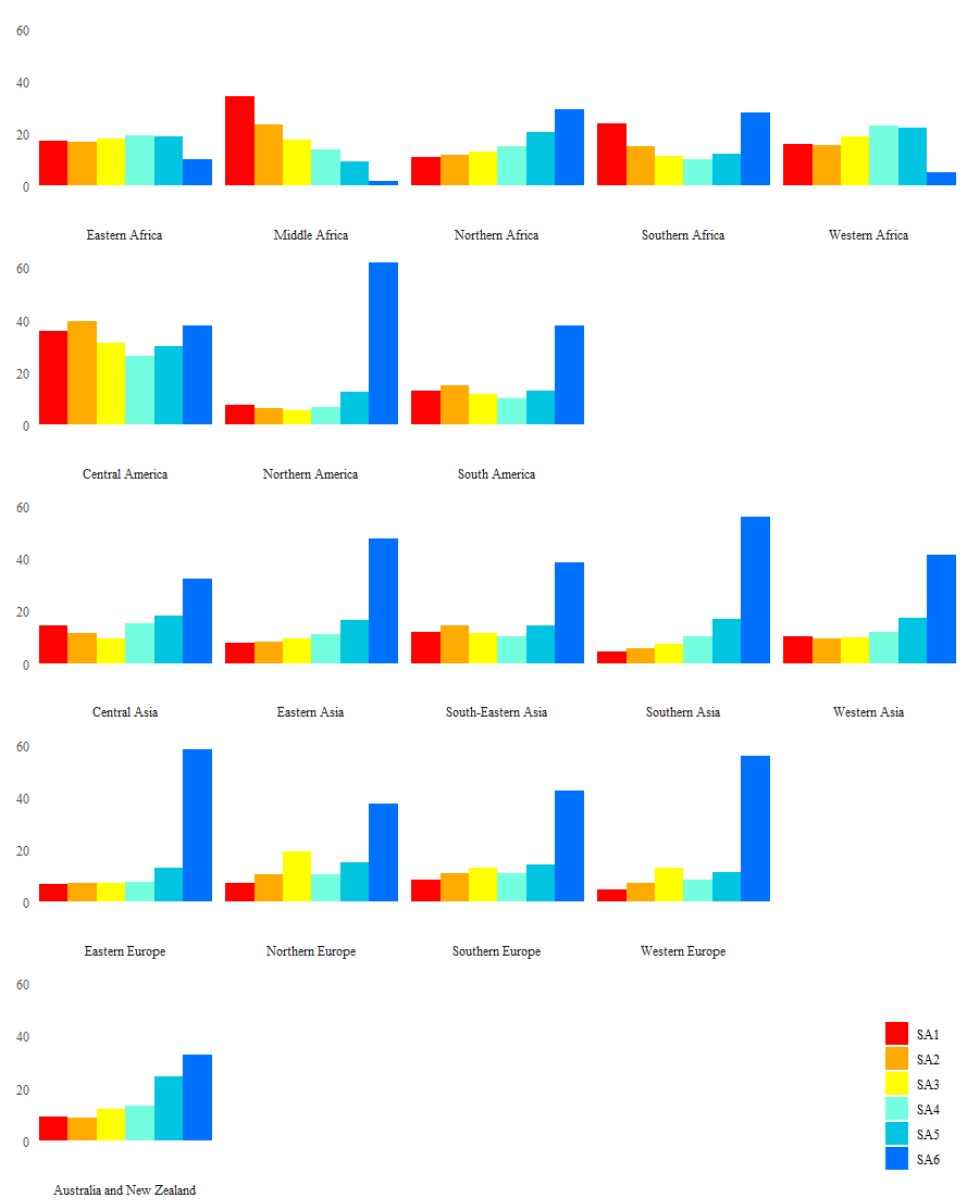




In terms of high $SA_6$ contributions to the regions identified above, it is likely that the underlying land cover products were mapping *temporary crops* or *arable land* at country level rather than cropland area, as also suggested by Tubiello et al. (2023b). These are in fact the specific sub-components of cropland included in all their definitions. Conversely, low $SA_6$ shares point to complex and fragmented agricultural landscapes in specific regions, where land cover products are likely to disagree. Indeed, we tested a possible relation between relative uncertainty in regional cropland area estimates and the level of contribution of top agreement classes (combined area of $SA_4$, $SA_5$ and $SA_6$) and found good correlation between the area uncertainty and the percent contribution to cropland area of the top three agreement classes (Fig. 5).

**Figure 5.** Linear regression of relative uncertainty in regional cropland area estimates against percent contribution to same cropland area by top-three agreement classes ($R^2$=0.46, NRMSE=53%).

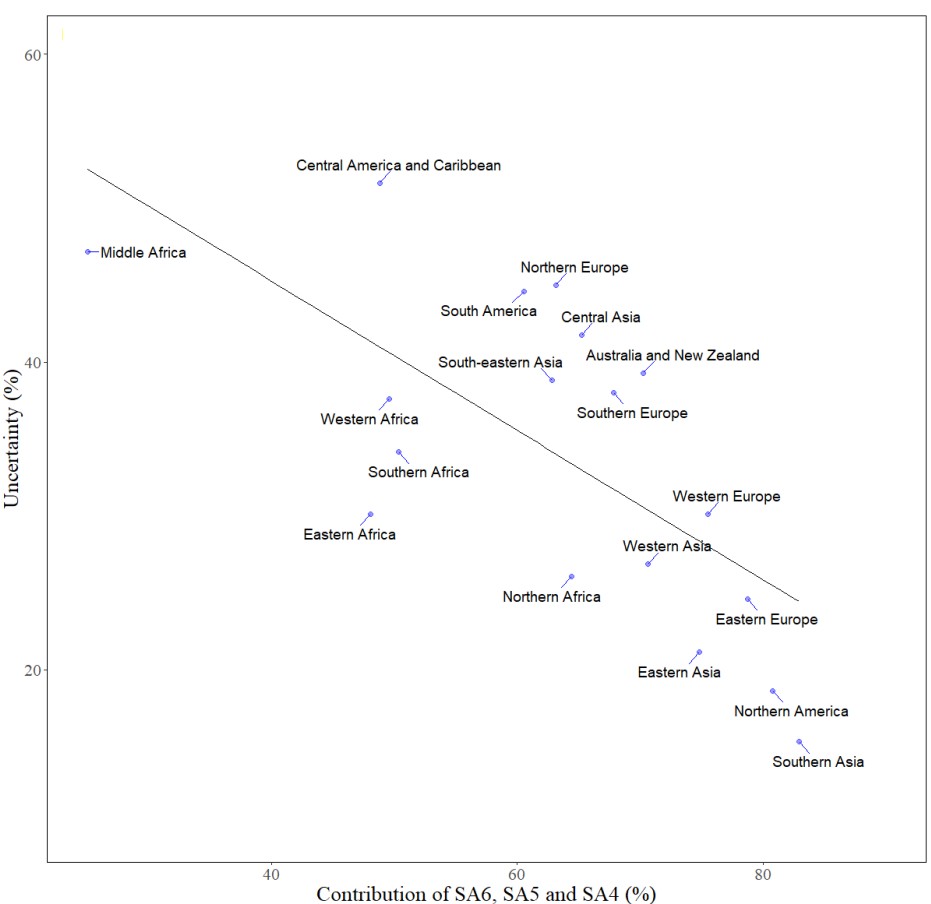



### 3.2.2 Country results


We extended the above analysis based on simple agreement classes to countries. The CAM country data confirmed the regional analysis of a strong link between simple cropland landscapes and prevalence of the $SA_6$ contribution to total cropland area. Only five countries globally had $SA_6 > 65\%$, of which four in Eastern Europe: Ukraine (75%), Bulgaria (70%), Hungary (70%), the Republic of Moldova (67%) and


Canada (67%). Cropland in these countries is indeed dominated by large, easy-to-recognize-from-space agricultural fields of annual crops (FAO, 2023a). In Ukraine, the top three agreement classes contributed 90% of the cropland area. Virtually the same features applied to other countries in Eastern and Central Europe, specifically Czechia (64%), Slovakia (64%), Romania (63%), Serbia (62%), Germany (62%) and Poland (61%); as well as in France and Austria, where it exceeded 50%.


Outside of Europe, Canada and the USA also had substantial proportions of $SA_6$ in their cropland areas, accounting for 67% and 61% respectively. In Central Asia, Turkmenistan and Uzbekistan show values comparable to Western European countries, with 52% and 57% respectively. In South eastern Asia, large agreement in cropland classification is found in India (58%), Pakistan (63%), Bangladesh (56%), and Thailand (57%). Among African countries, Egypt stood out as the only one with a significant share from


$SA6$ (62%), likely due to the presence of the irrigated fields along the Nile and of pivot irrigation schemes against an otherwise arid landscape, which were well captured by the six classification algorithms.

Similarly and consistently with the regional findings, the CAM country data likewise suggested a relation between complexity of cropland landscapes and low $SA_6$ contribution. Indeed, of the 17 countries with $SA_6$ contribution below 10% and cropland size above 5 Mha (threshold chosen arbitrarily), 14 were


located in Eastern, Middle or Western Africa, including Burkina Faso, Niger and DRC (0% $SA_6$ contribution, implying no agreement across the six maps in any pixel), Mozambique, Mali and Uganda (1%).

Finally, the highest uncertainties in estimated cropland area corresponded to high disagreement of the underlying land cover maps, expressed herein as the fraction covered by $SA_1 > 80\%$. These country cases


included Papua New Guinea and Sierra Leone ($SA_1=90\%$)(Appendix C, Tab. C2). The difficulties to map the fragmented and heterogeneous agricultural landscapes that prevail in these countries likely contributed to this feature (Potapov et al., 2022). In addition, in Papua New Guinea, cropland is dominated by permanent crops (FAO, 2023a), which most CAM input layers do not include in their definitions. This means that in such countries, CAM is mapping arable land rather than cropland per se, i.e., there is bias in the presence


of permanent crops.



Similar yet more complex dynamics were associated to country cases with 100% uncertainty in the CAM cropland estimates (Fig. 6). Two typologies could be identified among countries with cropland area > 1 Mha (Appendix C, Tab. C2). The first country case was characterized by high $SA_1$ percent contribution to total cropland area, and included Nicaragua (50%), DRC (49%) and Malaysia (36%). As in the previous cases, dominance of the $SA_1$ class was linked to complex landscapes within cropland, which could not be mapped precisely, leading to high uncertainty. In particular, Malaysia was characterized by a large presence of permanent crops, which could not be mapped by all products. The second typology was characterized by high $SA_3$ percent contributions to total cropland area and included Ireland (54%) and New Zealand (36%). In both cases, despite little presence of permanent crops, disagreement across land cover products persisted, but within arable land. We speculated that $SA_3$ prevalence was linked to presence of crop/pasture mixtures within cropland, in fact large shares of temporary meadows and pastures within arable land—a well-known landscape in both countries—which are mapped only by a subset of the underlying land cover products, generating high uncertainty as in the previous case, but for different reasons.

**Figure 6. Contribution to cropland area by cropland agreement class in 5 countries with 100% uncertainty and cropland area > 1Mha.**

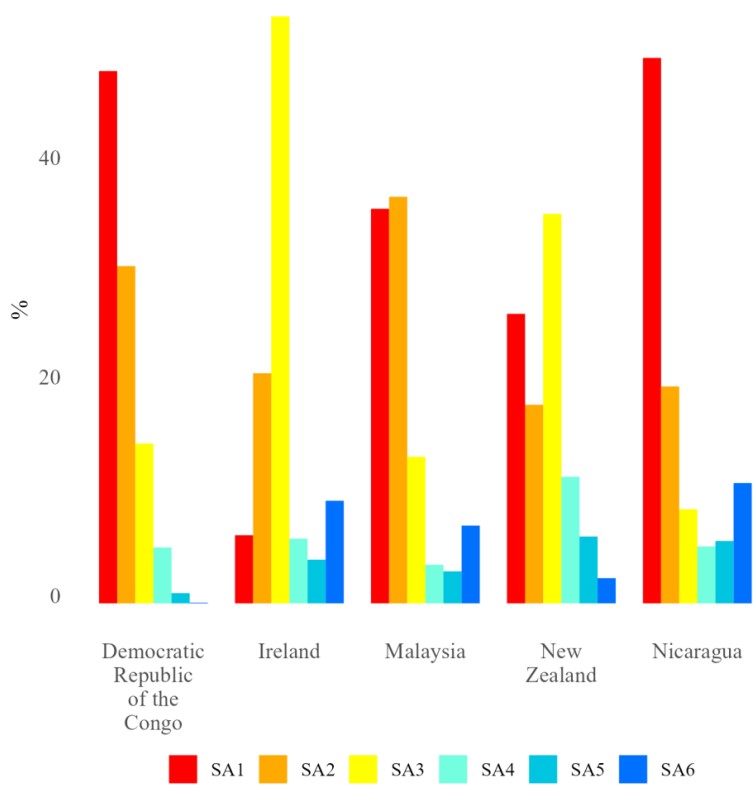

## 3.3 Detailed cropland agreement

### 3.3.1 Regional results

To gain further insight into the relationship between mapping uncertainty and landscape complexity, we looked at the detailed contribution of single CAM land cover maps to $SA_1$ contributions, by region, where $SA_1$ represents the areas with minimum agreement across land cover products. The detailed combinations at the level of the regions discussed above is available in the CAM dataset (Tubiello et al., 2023a) but too large to be discussed herein in its entirety. We limited the analysis herein to the larger FAO regional groups and included a discussion of notable country examples among those discussed earlier.

At global level, consistently with their cropland definitions (Tab. 1), and with reference to Fig. 7, GLAD contributed less than 1% in Europe to about 5% in Africa; WorldCover 1% in the Americas, Europe and Oceania to 9% in Africa; ESRI 4% in Europe to 11% in Oceania. In fact, contributions to $SA_1$ were largely



from FCS30 (23% in Africa to 60% in Oceania); Globeland30 (23% in Oceania to 53% in Europe); and

FROM_GLC (30% in Africa to 8% in Oceania).

**Figure 7.** Percent contribution to minimum cropland agreement (SA1), by input land cover product used in CAM (ESR=ESRI; FCS30=GLC_FCS30-2020; FRG=FROM_GLC Plus; GLD=GLAD; GL30=Globeland30; WCO=WorldCover).

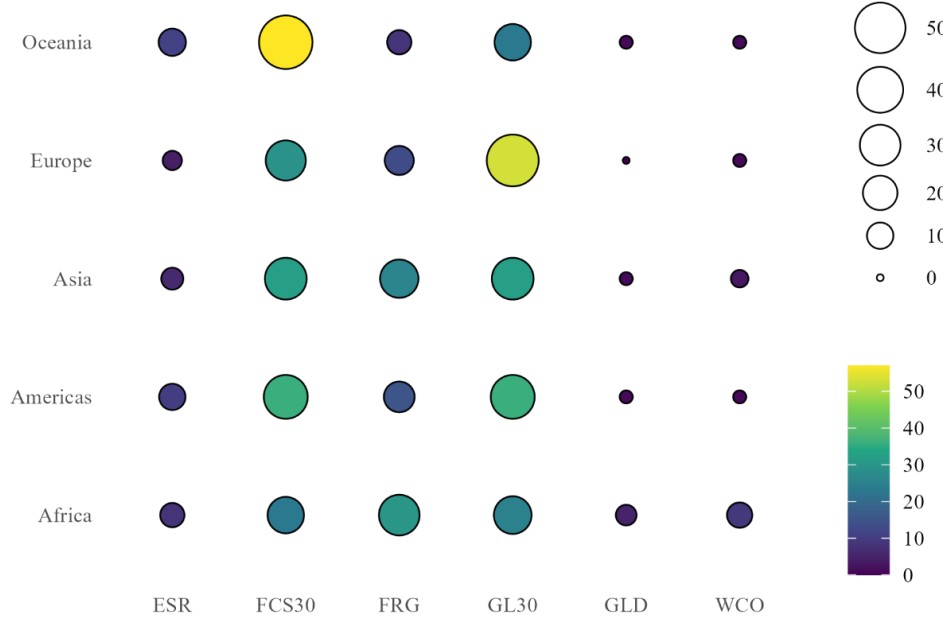

The regional analysis of detailed agreement singles out FCS30 and Globeland30 as the land cover products
in CAM with the largest contribution to disagreement across regions, consistently with the fact that these
are the only maps that included permanent crops in their definitions. Conversely ESRI, GLAD and
WorldCover were the least contributors to disagreement, in line with their definitions which focus on
herbaceous crops within cropland landscapes. The FROM_GLC was an intermediate case, consistently with
its inclusion of shrub crops within its definitions.

### 3.3.2 Country results

In our earlier observations, we highlighted that Ireland and New Zealand had the largest contribution of
SA$_3$, accounting for 54% and 36% respectively. This indicated that, on average, half of the land cover
products in CAM agreed on mapping cropland. The detailed cropland agreement data in CAM showed that
in both countries, this outcome was due to a fixed combination of just three products: ESRI-FCS30-
Globeland30 (Fig. 8). These were in fact the only land cover products in CAM that include pastures or





generic herbaceous cover within their cropland definition, confirming our hypothesis that prevalence of $SA_3$ was linked to extensive areas of pastures within cropland, indeed typical of both countries' agricultural landscapes. Additionally, for New Zealand, the detailed agreement data indicated that FSC30 and FSC30-

GL30 were behind the $SA_1$ and $SA_2$ contributions to cropland area in the country. This was consistent with a significant presence of permanent crops in the national agricultural landscape, as both products were the only ones in CAM that could capture *permanent crops* within cropland area.

**Figure 8.** Percent contribution of products combinations to the detailed agreement, in Ireland and New Zealand. (Detailed agreement limited to land cover combinations with at least 3% contribution. ESR=ESRI;

FCS30=GLC_FCS30-2020; FRG=FROM_GLC Plus; GLD=GLAD; GL30=Globeland30; WCO=WorldCover).

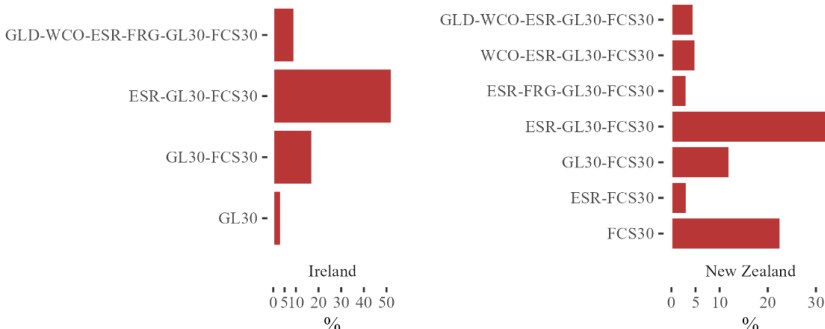

For DRC, Malaysia, and Nicaragua (Fig. 9), countries characterized by cropland area estimates with very high uncertainty and a dominance of the $SA_1$ class, the information provided by the detailed agreement data

in CAM indicated that FCS30, GL30 and FRG were the land cover products mainly contributing to $SA_1$. As discussed above, FCS30 and GL30 were the only two products in CAM that included permanent crops in their definitions. They compared in 8 of nine $SA_1$ combinations in DRC, 7 out of 8 in Malaysia, and in 5 out of 6 combinations shown in Fig. 9, consistently with the large presence of *permanent crops* in these countries.






**Figure 9.** Percent contribution of products combinations to the detailed agreement, in the Democratic Republic of the Congo, Malaysia and Nicaragua. (Detailed agreement limited to land cover combinations with at least 3% contribution. ESR=ESRI; FCS30=GLC_FCS30-2020; FRG=FROM_GLC Plus; GLD=GLAD; GL30=Globeland30; WCO=WorldCover).

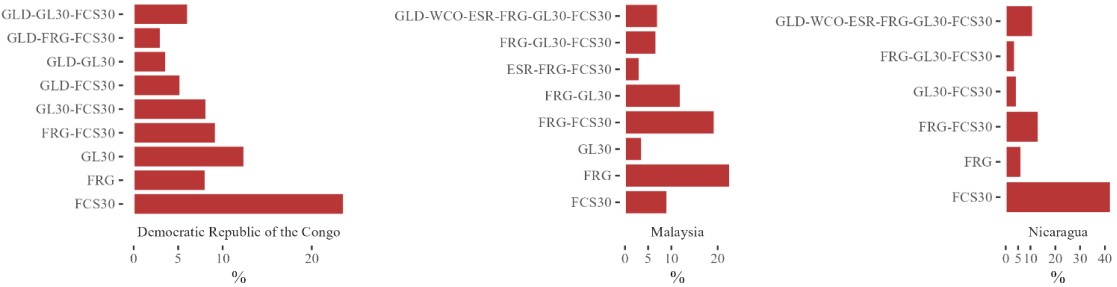


## 4. Data availability

The CAM dataset is publicly available in Zenodo at: https://doi.org/10.5281/zenodo.7987515 (Tubiello et al., 2023a).

## 5. Conclusions

The CAM dataset presented herein consolidates information from six high-resolution global cropland maps circa 2020 currently available in the literature, using a meta-analysis approach to estimate cropland area and its uncertainty at country level, with data for 221 countries and territories and 34 regional aggregates. The CAM data are complemented by ancillary data on simple and detailed agreements of the underlying land cover products, with the same country and regional coverage. To our knowledge, this is the first time

that such country information is presented in the literature.

The global regional and country examples provided in this work demonstrate the usefulness of the CAM dataset to assess current knowledge on cropland area in countries as available from land cover maps, highlighting how they agree or disagree on specific agricultural landscapes, depending on individual accuracy but also and importantly on definitional differences. In particular, the data highlighted critical

connections between the level of complexity in the observed agricultural landscape and the preponderance of specific cropland agreement classes. We showed that high agreement among land cover products corresponded to large scale fields with high input annual crops, hence cropland areas dominated by *temporary crops*; while minimum agreement tended to correspond to presence of more complex cropland landscapes, be it tree plantations in Africa, South-eastern Asia and South America, or mixed crop-pasture

systems, such as in Ireland and New Zealand.



The CAM dataset represents a new global knowledge product and can serve as useful guide to support future land cover and land use product development and data evaluation.





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

## Authors contribution

FNT GC and LC conceived the dataset and conducted the analysis. GC, LC, PH and ZC supported the underlying geospatial analysis. GD supported the land use analysis and country examples. All authors contributed to writing the manuscript.

**Competing interests**

At least one of the (co-)authors is a member of the editorial board of Earth System Science Data.

## Acknowledgements

The authors wish to acknowledge the Swiss Office of Agriculture for funding the FAO Statistics Division under the FAO Regular Programme, which made this work possible. The views expressed in this work
are the authors' only and cannot be taken to represent FAO's views or policy on the subject matter.



# 8. Appendices

## Appendix A. The six land cover products of CAM maps and measures of spatial consistency and similarity


The six cropland layers that contribute to CAM include one thematic cropland product and five global land cover products, all containing one or multiple cropland classes (Tab. A1). GLAD, the thematic cropland map, uses images of multiple years (2016–2019) to create a single cropland map; a similar approach is used by FROM_GLC, Globeland30 (4 years) and the FCS30 (3 years) while ESRI and WorldCover use 2020

images. ESRI and Globeland30 make use of a combination of pixel- and object-based classification methods whereas the other products use pixel-based supervised classification approaches. Of the CAM components, WorldCover was openly available from the Earth Engine Data Catalog, while the other datasets were available through assets created by individual users in the GEE environment.

**Table A1. Technical specifications of six land cover maps used as input in CAM maps (Tubiello et al., 2022; Tubiello et al., 2023b) and underlying information in the CAM dataset (Tubiello et al., 2023a).**

| Cropland layer | Spatial Resolution | Remote sensing data | Classification method | Algorithm | Source |
|---|---|---|---|---|---|
| **ESRI** | 10m | Sentinel-2 | Pixel & Object | Convolutional Neural Network | Karra et al., 2021 |
| **FROM_GLC Plus** | 30m | Landsat OLI & ETM+ MODIS | Pixel | Random Forest | Yu et al., 2022 |
| **GLAD** | 30m | Landsat Analysis Ready Data (ARD) | Pixel | Bagged Decision Tree Ensemble | Potapov et al., 2022a |
| **GLC-FCS30-2020 (FCS30)** | 30m | Landsat, Sentinel-1SAR, SRTM DEM | Pixel | Local adaptive Random Forest | Zhang et al., 2021 |
| **GLOBELAND30** | 30m | Landsat TM5, ETM+ & OLI, HJ-1, GF-1 | Pixel & Object (POK[a]) | Pixel-Object-Knowledge Classifier | Chen et al., 2015 |
| **WORLDCOVER** | 10m | Sentinel-1 & 2, Copernicus Global DEM, RESOLVE Ecoregions 2017 | Pixel & Position | Gradient Boosting Decision Tree Algorithm | Zanaga et al., 2021 |

[a]Pixel- and object-based methods with prior knowledge.




Spatial consistency and similarity of the six input layers in CAM was investigated following methods in Liu et al. (2021). A total of 30,000 random points among non-zero cropland values were selected, and cropland area fractions for each agreement layer separately, for all pixels within an area of 5 x 5 km around each random point. Areas of overlap were excluded from the analysis. This produced approximately 28,000

data points for each layer containing the location and cropland area fraction. Scatter plots were created where the cropland area fraction was plotted for each dataset pair. This allowed for a comprehensive analysis of the spatial similarity among datasets. The pixel-level comparison yielded the best agreement between the GLAD and the WorldCover ($R^2 = 0.79$; RMSE = 0.15), followed by Worldcover and ESRI ($R^2 = 0.61$, RMSE = 0.23), and ESRI and GLAD ($R^2 = 0.58$, RMSE = 0.24). On the contrary, lowest $R^2$ is found

between FROM_GLC and the Globeland30 ($R^2 = 0.26$, RMSE = 0.36) and FROM_GLC and FCS30 ($R^2 = 0.33$, RMSE = 0.32) even though in absolute terms these products all mapped the largest global extents (Tab. A2). Overall, there is a clear separation between the three products that correlate well with one another (Worldcover, GLAD and ESRI) and the other high-resolution products. On the other hand, both Globeland30 and FCS30 have values that are almost consistently higher than in the GLAD. This suggests

that there are many regions which are classified as cropland for Globeland30 and FCS30 where GLAD, but also WorldCover and ESRI, show no presence for cropland (Fig. A1 for pixel-comparison of the GLAD with the other six land cover maps).

**Table A2. Pixel-level correlation ($R^2$) between the 6 cropland datasets, RMSE in brackets (values computed from fraction of the cropland area).**

|             | ESRI        | FROM_ GLC   | GLAD        | FCS30       | Globeland30 |
| ----------- | ----------- | ----------- | ----------- | ----------- | ----------- |
| FROM_GLC    | 0.4 (0.29)  |             |             |             |             |
| GLAD        | 0.58 (0.24) | 0.5 (0.25)  |             |             |             |
| FCS30       | 0.45 (0.3)  | 0.33 (0.32) | 0.48 (0.31) |             |             |
| Globeland30 | 0.47 (0.32) | 0.26 (0.36) | 0.46 (0.33) | 0.39 (0.27) |             |
| Worldcover  | 0.61 (0.23) | 0.53 (0.24) | 0.79 (0.15) | 0.49 (0.3)  | 0.47 (0.32) |




**Figure A1. Density scatterplot of correlation between GLAD and the other 5 cropland datasets. The colour of the points indicates the number of scatter points in that location. The black line depicts the regression line, while the red line shows the optimal 1:1 relationship.**

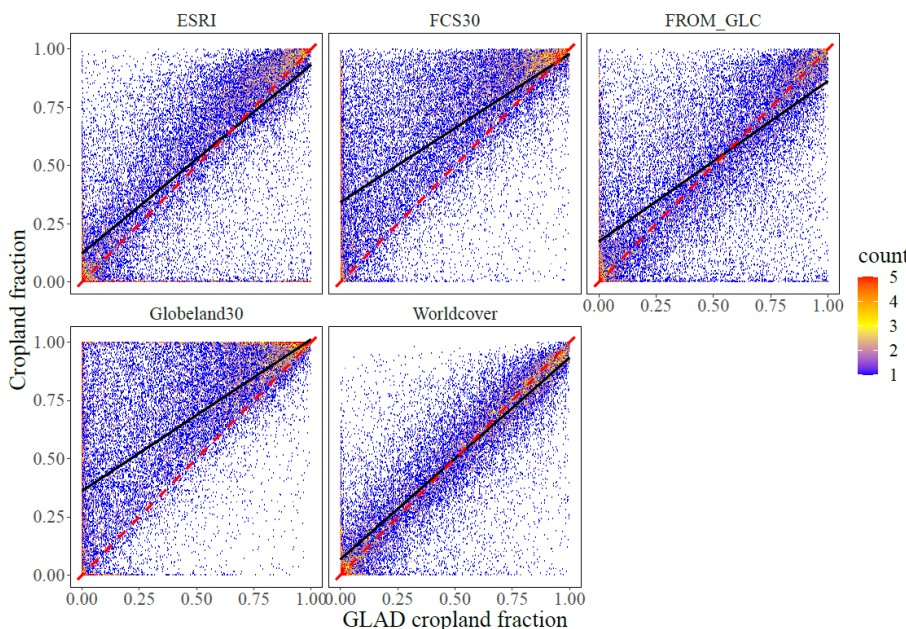


Binary similarity and distance measures are helpful tools in analysis of patterns and clustering (Choi et al., 2009). The detailed agreement allows to extract information on the binary instances between two cropland layers and to define for each country their intersections and their mismatches. We computed the Baroni similarity index (Baroni-Urbani and Buser, 1976) from country statistics of the detailed agreement. The

Baroni similarity index ranges between 0 (no attributes in common between pairs of land cover products) and 1 (perfect overlap) and it thus accounts for both 'positive' – that is, where two layers agree on the presence of cropland—and 'negative' matches, corresponding herein to areas where two layers agree on the absence of cropland. The normalised index was computed as follows:

$$Similarity\ index_{ij} = \frac{\sqrt{AD} + A}{\sqrt{AD} + A + B + C} \qquad (1)$$

Where $i$ and $j$ are cropland layers; A is the area that both layers mapped as cropland; B is area of cropland mapped by the first layer only; C is the area of cropland mapped by the second layer only; D is the country area that both layers agree is not cropland. In the analysis, the total A + B + C + D corresponds to the total land area (see Tab. A3 for regional and global results).



**Table A3. Index of similarity, by region and globally.**

|  | Africa | Americas | Asia | Europe | Oceania | World |
|---|---|---|---|---|---|---|
| ESRI_FROM_GLC | 0.67 | 0.82 | 0.80 | 0.85 | 0.79 | 0.80 |
| ESRI_GLB30 | 0.70 | 0.85 | 0.81 | 0.86 | 0.89 | 0.83 |
| FCS30_ESRI | 0.66 | 0.83 | 0.80 | 0.88 | 0.85 | 0.82 |
| FCS30_FROM_GLC | 0.75 | 0.78 | 0.82 | 0.82 | 0.73 | 0.80 |
| FCS30_GLB30 | 0.75 | 0.84 | 0.82 | 0.87 | 0.88 | 0.83 |
| GLB30_FROM_GLC | 0.72 | 0.76 | 0.80 | 0.80 | 0.77 | 0.78 |
| GLD_ESRI | 0.72 | 0.89 | 0.86 | 0.89 | 0.89 | 0.86 |
| GLD_FCS30 | 0.79 | 0.81 | 0.80 | 0.84 | 0.86 | 0.82 |
| GLD_FROM_GLC | 0.75 | 0.88 | 0.81 | 0.89 | 0.85 | 0.84 |
| GLD_GLB30 | 0.81 | 0.82 | 0.81 | 0.81 | 0.89 | 0.82 |
| GLD_WCO | 0.83 | 0.92 | 0.89 | 0.93 | 0.92 | 0.90 |
| WCO_ESRI | 0.73 | 0.89 | 0.86 | 0.88 | 0.88 | 0.86 |
| WCO_FCS30 | 0.76 | 0.82 | 0.82 | 0.84 | 0.81 | 0.82 |
| WCO_FROM_GLC | 0.75 | 0.86 | 0.81 | 0.87 | 0.87 | 0.83 |
| WCO_GLB30 | 0.78 | 0.81 | 0.82 | 0.81 | 0.85 | 0.81 |
| Average | 0.74 | 0.84 | 0.82 | 0.86 | 0.85 | 0.83 |










## Appendix B. Binary and decimal code attributes of the detailed map of agreement

Table B1. Lookup table of the binary and decimal values for each detailed class of agreement. For each datasets a value of 1 means presence of cropland and 0 is absence.

| Value | | Datasets | | | | | |
|---|---|---|---|---|---|---|---|
| Binary | Decimal | GLAD | WorldCover | ESRI | FROM_GLC | Globeland30 | FCS30 |
| 000000 | 0 | 0 | 0 | 0 | 0 | 0 | 0 |
| 000001 | 1 | 0 | 0 | 0 | 0 | 0 | 1 |
| 000010 | 2 | 0 | 0 | 0 | 0 | 1 | 0 |
| 000011 | 3 | 0 | 0 | 0 | 0 | 1 | 1 |
| 000100 | 4 | 0 | 0 | 0 | 1 | 0 | 0 |
| 000101 | 5 | 0 | 0 | 0 | 1 | 0 | 1 |
| 000110 | 6 | 0 | 0 | 0 | 1 | 1 | 0 |
| 000111 | 7 | 0 | 0 | 0 | 1 | 1 | 1 |
| 001000 | 8 | 0 | 0 | 1 | 0 | 0 | 0 |
| 001001 | 9 | 0 | 0 | 1 | 0 | 0 | 1 |
| 001010 | 10 | 0 | 0 | 1 | 0 | 1 | 0 |
| 001011 | 11 | 0 | 0 | 1 | 0 | 1 | 1 |
| 001100 | 12 | 0 | 0 | 1 | 1 | 0 | 0 |
| 001101 | 13 | 0 | 0 | 1 | 1 | 0 | 1 |
| 001110 | 14 | 0 | 0 | 1 | 1 | 1 | 0 |
| 001111 | 15 | 0 | 0 | 1 | 1 | 1 | 1 |
| 010000 | 16 | 0 | 1 | 0 | 0 | 0 | 0 |
| 010001 | 17 | 0 | 1 | 0 | 0 | 0 | 1 |
| 010010 | 18 | 0 | 1 | 0 | 0 | 1 | 0 |
| 010011 | 19 | 0 | 1 | 0 | 0 | 1 | 1 |
| 010100 | 20 | 0 | 1 | 0 | 1 | 0 | 0 |
| 010101 | 21 | 0 | 1 | 0 | 1 | 0 | 1 |
| 010110 | 22 | 0 | 1 | 0 | 1 | 1 | 0 |
| 010111 | 23 | 0 | 1 | 0 | 1 | 1 | 1 |
| 011000 | 24 | 0 | 1 | 1 | 0 | 0 | 0 |
| 011001 | 25 | 0 | 1 | 1 | 0 | 0 | 1 |
| 011010 | 26 | 0 | 1 | 1 | 0 | 1 | 0 |
| 011011 | 27 | 0 | 1 | 1 | 0 | 1 | 1 |
| 011100 | 28 | 0 | 1 | 1 | 1 | 0 | 0 |
| 011101 | 29 | 0 | 1 | 1 | 1 | 0 | 1 |
| 011110 | 30 | 0 | 1 | 1 | 1 | 1 | 0 |
| 011111 | 31 | 0 | 1 | 1 | 1 | 1 | 1 |
| 100000 | 32 | 1 | 0 | 0 | 0 | 0 | 0 |
| 100001 | 33 | 1 | 0 | 0 | 0 | 0 | 1 |
| 100010 | 34 | 1 | 0 | 0 | 0 | 1 | 0 |





| | | | | | | |
|---|---|---|---|---|---|---|
| **100011** | 35 | 1 | 0 | 0 | 0 | 1 | 1 |
| **100100** | 36 | 1 | 0 | 0 | 1 | 0 | 0 |
| **100101** | 37 | 1 | 0 | 0 | 1 | 0 | 1 |
| **100110** | 38 | 1 | 0 | 0 | 1 | 1 | 0 |
| **100111** | 39 | 1 | 0 | 0 | 1 | 1 | 1 |
| **101000** | 40 | 1 | 0 | 1 | 0 | 0 | 0 |
| **101001** | 41 | 1 | 0 | 1 | 0 | 0 | 1 |
| **101010** | 42 | 1 | 0 | 1 | 0 | 1 | 0 |
| **101011** | 43 | 1 | 0 | 1 | 0 | 1 | 1 |
| **101100** | 44 | 1 | 0 | 1 | 1 | 0 | 0 |
| **101101** | 45 | 1 | 0 | 1 | 1 | 0 | 1 |
| **101110** | 46 | 1 | 0 | 1 | 1 | 1 | 0 |
| **101111** | 47 | 1 | 0 | 1 | 1 | 1 | 1 |
| **110000** | 48 | 1 | 1 | 0 | 0 | 0 | 0 |
| **110001** | 49 | 1 | 1 | 0 | 0 | 0 | 1 |
| **110010** | 50 | 1 | 1 | 0 | 0 | 1 | 0 |
| **110011** | 51 | 1 | 1 | 0 | 0 | 1 | 1 |
| **110100** | 52 | 1 | 1 | 0 | 1 | 0 | 0 |
| **110101** | 53 | 1 | 1 | 0 | 1 | 0 | 1 |
| **110110** | 54 | 1 | 1 | 0 | 1 | 1 | 0 |
| **110111** | 55 | 1 | 1 | 0 | 1 | 1 | 1 |
| **111000** | 56 | 1 | 1 | 1 | 0 | 0 | 0 |
| **111001** | 57 | 1 | 1 | 1 | 0 | 0 | 1 |
| **111010** | 58 | 1 | 1 | 1 | 0 | 1 | 0 |
| **111011** | 59 | 1 | 1 | 1 | 0 | 1 | 1 |
| **111100** | 60 | 1 | 1 | 1 | 1 | 0 | 0 |
| **111101** | 61 | 1 | 1 | 1 | 1 | 0 | 1 |
| **111110** | 62 | 1 | 1 | 1 | 1 | 1 | 0 |
| **111111** | 63 | 1 | 1 | 1 | 1 | 1 | 1 |

Note: since we only have six datasets only 6 bits are shown for the binary value as the last 2 are always 0 (8-bit integer). An example: pixel value 38 equals bit 100110 meaning the 2nd, 3rd, and 6th bit (or datasets) depicts the presence of cropland (i.e. Globeland30, FROM_GLC and GLAD), whereas the other 3 layers show no cropland.



## Appendix C. CAM estimates for subregions and countries with largest relative uncertainty

**Table C1.** Percent area contribution by agreement class to regional cropland area estimates in CAM.

| Region | $SA_6$ | $SA_5$ | $SA_4$ | $SA_3$ | $SA_2$ | $SA_1$ |
|---|---|---|---|---|---|---|
| Eastern Africa | 10% | 19% | 19% | 18% | 17% | 17% |
| Northern Africa | 29% | 20% | 15% | 13% | 12% | 11% |
| Southern Africa | 28% | 12% | 10% | 11% | 15% | 24% |
| Western Africa | 5% | 22% | 23% | 19% | 16% | 16% |
| Middle Africa | 10% | 19% | 19% | 18% | 17% | 17% |
| | | | | | | |
| Northern America | 62% | 13% | 6% | 5% | 6% | 8% |
| Central America and Caribbean | 25% | 13% | 11% | 14% | 19% | 18% |
| South America | 38% | 13% | 10% | 12% | 15% | 13% |
| | | | | | | |
| Central Asia | 32% | 18% | 15% | 9% | 11% | 14% |
| Eastern Asia | 47% | 16% | 11% | 9% | 8% | 8% |
| Southern Asia | 56% | 17% | 10% | 7% | 6% | 4% |
| South-eastern Asia | 38% | 14% | 10% | 11% | 14% | 12% |
| Western Asia | 41% | 17% | 12% | 10% | 9% | 10% |
| | | | | | | |
| Eastern Europe | 58% | 13% | 8% | 7% | 7% | 7% |
| Northern Europe | 38% | 15% | 11% | 19% | 10% | 7% |
| Southern Europe | 43% | 14% | 11% | 13% | 11% | 8% |
| Western Europe | 56% | 11% | 9% | 13% | 7% | 5% |
| | | | | | | |
| Australia and New Zealand | 33% | 24% | 13% | 12% | 9% | 9% |



**Table C2**. Countries with 100% relative uncertainty in CAM estimates (mean area; SE; area by simple agreement class) and FAO cropland area.

| Country | CAM | SE | SA1 | SA2 | SA3 | SA4 | SA5 | SA6 | FAO |
|---|---|---|---|---|---|---|---|---|---|
| | | | | | 1000 ha | | | | |
| Bahamas | 50 | 50 | 42 | 7 | 1 | 0 | 0 | 0 | 12 |
| Bhutan | 50 | 50 | 25 | 12 | 6 | 3 | 2 | 1 | 100 |
| Central African Republic | 500 | 500 | 356 | 92 | 37 | 13 | 2 | 0 | 1880 |
| Haiti | 200 | 200 | 89 | 43 | 26 | 20 | 14 | 8 | 1350 |
| Honduras | 700 | 700 | 287 | 156 | 105 | 55 | 43 | 54 | 1596 |
| Ireland | 2000 | 2000 | 124 | 420 | 1071 | 118 | 80 | 187 | 445 |
| Malaysia | 3000 | 3000 | 1080 | 1113 | 401 | 105 | 88 | 213 | 8286 |
| New Zealand | 4000 | 4000 | 1057 | 725 | 1421 | 462 | 244 | 92 | 601 |
| Nicaragua | 2000 | 2000 | 995 | 396 | 172 | 104 | 114 | 220 | 1790 |
| Panama | 600 | 600 | 213 | 184 | 81 | 41 | 40 | 40 | 665 |
| Papua New Guinea | 300 | 300 | 253 | 33 | 9 | 4 | 1 | 0 | 1000 |
| Timor-Leste | 200 | 200 | 113 | 47 | 17 | 11 | 8 | 4 | 191 |
| Puerto Rico | 70 | 70 | 30 | 17 | 10 | 6 | 4 | 2 | 65 |
| Qatar | 20 | 20 | 7 | 3 | 3 | 2 | 2 | 2 | 24 |
| Sierra Leone | 200 | 200 | 181 | 16 | 3 | 1 | 0 | 0 | 1749 |
| Western Sahara | 10 | 10 | 8 | 2 | 0 | 0 | 0 | 0 | 4 |
| Suriname | 60 | 60 | 18 | 9 | 6 | 10 | 13 | 4 | 67 |
| Eswatini | 200 | 200 | 55 | 44 | 26 | 16 | 18 | 41 | 190 |
| Trinidad and Tobago | 20 | 20 | 10 | 5 | 3 | 1 | 1 | 0 | 47 |
| United Arab Emirates | 100 | 100 | 43 | 28 | 15 | 6 | 4 | 3 | 90 |
| Democratic Republic of the Congo | 5000 | 5000 | 2428 | 1538 | 729 | 255 | 46 | 3 | 15372 |
| Montenegro | 90 | 90 | 51 | 18 | 9 | 6 | 4 | 2 | 15 |