# Peer review of "A new cropland area database by country circa 2020"

_Earth System Science Data, 2023_

## Author Comment (AC1)

Reviewer 1:

Summary

Reliable mapping cropland area is essential for assessing and monitoring the sustainability of agriculture from regional to global scale. Here Tubiello et al., generated country-level crop area dataset based on six remote sensing-derived crop area dataset at the global scale. The generated dataset agreed well with FAO land use statistics; with the generated dataset, the authors conducted a comprehensive analysis regarding the crop area and its uncertainty. The manuscript is generally well organized, and the research is important. Here, I listed a few concerns regarding the manuscript.

Specific comments

We thank the reviewer for the positive comment. Kindly find our answers below.

1)    Line 47-49, since there have been cropland agreement maps in the previous study, is it suitable to refer to the new dataset in this study as "Cropland Agreement Mapping dataset"? It seems that the name did not reflect the difference between the dataset in this study and that of previous study. My understanding is that one major difference between the two datasets is that this study aggregated the previous dataset to the country level, correct?

Right, we will use the term "Cropland Agreement Mapping dataset" to refer to this new analysis.

2)    Line 50, 25% of what? I cannot understand this sentence.

We will clarify the sentence as follows "...area can be estimated to within 25% of the mean cropland area."

3)    Line 86-87, actually, the R2 and MAE between the crop area of CAM and FAO can be compared with the performance between the crop area of a single crop data (one of the six crop area data) and FAO.

Indeed, we made extensive analysis of the correspondence of each dataset with FAOSTAT statistics in our preparatory work. However, we decided not to include that analysis in this paper to avoid duplications as some of the six input layers (e.g. GLAD) already published a comparison with FAOSTAT statistics at regional level. Instead, we focused on the comparison of our synthetic product with FAOSTAT statistics at country and regional level which represents novel information. We futher discuss the R2 and NRMSE under point 9.

4)    Line 118-119, how did you aggregate the pixel-level information to national scale? What kind of coordinate system is used for the dataset? Did you consider the area differences across different grid cells or pixels? Please clarify it.

The pixel level data was aggregated into country level information using the cloud processing Google Earth Engine platform. Here an image is created where pixel values correspond to the area of the pixel. By construction, GEE processing takes into account the difference in area due to the latitude. The pixels of this image are summed together, grouped by agreement class, to arrive at the country level aggregates. So, the aggregation did take into account the difference in pixel area at higher latitudes. The dataset is produced using the WGS84 (EPSG:4326). We will add this information on the aggregation method and coordinate system used to the updated manuscript. Line 77: Coordinate system, Line 116-118: elaboration on aggregation method.

5) Line 127, "Conversely" or "Similarly"?

Indeed, Similarly is the right adverb – we will change it in the revised manuscript.

6) Line 131, "definitional bias would be higher for SAk classes" not clear. Here you mean "definitional bias would be higher for SAk classes with lower k values" correct? Please clarify it.

Correct, we will clarify this sentence.

7) Line 153, Pearson correlation coefficient is R instead of R2, here you mean R square?

The R2 should indeed be the coefficient of determination. We will specify this in the new version of the manuscript.

8) Line 171-174, it's hard to follow this part. I assume many readers would have problems similar to me. What's relative uncertainty, what does "communicate" here mean, is it a commonly used mathematical term? Here "dimension" corresponds to number of samples? Please clarify those terms or definitions and use plain language to explain what does this part exactly mean.

We have revised this paragraph to enhance readability. To answer the reviewer on the specifics point above—which have been incorporated in the revision—all terms used are accepted terminology in metrology, which is the branch of science/mathematics that has contributed to the currently accepted definitions of uncertainty, as published by the International Bureau of Weights and Measures (BIPM). Furhtemore, ''Dimension'' referred to sample size (revised); ''relative uncertainty'' is commonly understood as the uncertainty of a given measurand divided by its expected mean value, for instance in this case relative standard error. The overall important issue underlined in this paragraph is that one should not be carried away with precision where little exist. There are mathematical formulas that assess the ''uncertainty'' of the ''uncertainty'', when the latter is derived as estimated standard deviations from sample sizes of dimension n. This in turn limits the number of significant figures with which the latter can be expressed. See for instance: https://www.webassign.net/question_assets/unccolphysmechl1/measurements/manual.html (table following formula #38). We quote more precisely the Guide on Uncertainty of the BIPM in the revised text.

9)    Fig. 2-3, what does the red dashed line and black line represent, respectively? Please clarify it in the figure caption. Will the R2 between the crop area of CAM and FOASTAT be higher than that between the crop area of a single data (one of the six crop area data)? Will the    NRME    be    lower    by    using    multiple    data    ensemble    mean?

We will add to captions of Figure 2 and 3 the meaning of the red and black lines. R2 values are relatively similar between the different datasets as these are highly influenced by the large countries. However, for NRMSE we find that the CAM dataset provides considerably smaller errors than the individual datasets when compared to FAOSTAT cropland areas.

10)   Fig.4, the label and unit of y-axis should be given

We will add a label including units to the y-axis of Fig. 4.

11)    Fig. 5, is this regression line statistically significant? Please show the p-value. We will add the P-value to the caption of Figure 5.

12)   Fig.7, please add necessary description in the figure caption to explain what does the size of the circle and color respectively mean.

We    will    modify    Figure    7    into    a    more    readable    plot.

13)   In addition to the results, brief discussion about how the findings could aid further development of land cover or crop area products could potentially make this study more influential.

We agree that a discussion on the use of the results from this study for future land cover and cropland area products would be beneficial. We will add an extra section on the potential uses of the CAM dataset and of the Cropland Agreement Map for future work on land cover and cropland products in the revised manuscript.

Replies to reviewers 1 and 2 for essd-2023-211: A new cropland area database by country circa 2020

Reviewer 2:

Tubiello et al., have presented an important contribution to the field through the development of a novel cropland area mapping dataset (CAM). This dataset, which aggregates information from six high-resolution remote sensing-based global cropland maps, includes an estimation of uncertainty at country and regional levels. The systematic analysis of agreement and disagreement among the six contributing maps has provided valuable insights into the underlying uncertainties associated not only with individual map accuracies but also with their definitional differences.

The manuscript is generally well-written and presents the data in a clear and effective manner. The identification of regions with large uncertainty could prove particularly useful for future endeavors in land cover and land use product development and data evaluation.

We are greateful tot he reviewer for the positive evalution. Please see below detailed answers to the specific comments.

Nonetheless, there are a few points that could use further clarification:

Line 153: It is mentioned that R2 is the Pearson correlation coefficient, which seems to be a misunderstanding. R2 is the coefficient of determination.

Indeed, we will specify and clarify it in the new version of the manuscript.

Figure 1: I noticed a large area of low agreement in the Southeastern United States. It would be beneficial to readers if you could provide a more in-depth discussion about this. Specifically, what are the differences in how each of the six datasets defines 'cropland' in these areas? Are these differences primarily due to methodological variations or could there be other underlying factors contributing to the disparity?

The high spatial detail of the CAM map indeed allows for many interesting analyses and investigations of the pattern of agreement and disagreement beyond the country level presented by the CAM dataset and we are also very interested to study further these patterns. However, we thought that such specific analysis would add too much detail and complexity to this paper. We will add to the revised version of the manuscript a new paragraph with potential applications of the CAM dataset and map where we will discuss the potentials for this type of research. To answer on the specific case of Southeastern USA, we think the discrepancies (and therefore the lower agreement) are likely due to the areas of grasslands (pastures) and permanent crops of Florida that only some of the 6 land cover layers include in their operational definitions.

Additionally, it would be helpful if you could discuss the limitations of your approach, including any potential biases and how they might be addressed in future research.

We agree that a discussion on limitations of our approach would be beneficial. We will add an extra section to describe the limitations of our study and suggested future research for improvements in the revised manuscript.

I hope these comments help to further strengthen the manuscript.